# Baculovirus Display of Peptides and Proteins for Medical Applications

**DOI:** 10.3390/v15020411

**Published:** 2023-02-01

**Authors:** Aaron Pavel Rodríguez-Hernández, Daniel Martínez-Flores, Adolfo Cruz-Reséndiz, Teresa Padilla-Flores, Rodrigo González-Flores, Kenia Estrada, Alicia Sampieri, Aldo Román Camacho-Zarco, Luis Vaca

**Affiliations:** 1Instituto de Fisiología Celular, Universidad Nacional Autónoma de México, Ciudad Universitaria, Mexico City 04510, Mexico; 2Departamento de Microbiología y Parasitología, Facultad de Medicina, Universidad Nacional Autónoma de México, Ciudad Universitaria, Mexico City 04510, Mexico; 3European Molecular Biology Laboratory, Grenoble Outstation, EMBL Grenoble, CEDEX 9, 38042 Grenoble, France

**Keywords:** baculovirus display, GP64, VP39, vaccines

## Abstract

Baculoviridae is a large family of arthropod-infective viruses. Recombinant baculoviruses have many applications, the best known is as a system for large scale protein production in combination with insect cell cultures. More recently recombinant baculoviruses have been utilized for the display of proteins of interest with applications in medicine. In the present review we analyze the different strategies for the display of proteins and peptides on the surface of recombinant baculoviruses and provide some examples of the different proteins displayed. We analyze briefly the commercially available systems for recombinant baculovirus production and display and discuss the future of this emerging and powerful technology.

## 1. Introduction

Baculoviridae is a large family of arthropod-infective viruses [1]. With hundreds of members, only a handful of them have been studied over the last few years [1]. *Autographa californica* multiple nucleopolyhedrovirus (AcMNPV) is one of the best studied and characterized baculovirus to this date [2]. This virus is used in the baculovirus/insect cell expression system, utilized in the production of many commercially available vaccines, hormones, and bioactive peptides [3].

Baculoviruses have a genome consisting of a double-stranded DNA, which is why they are indexed in group I of the Baltimore Classification [4]. The circular genome size ranges from 80–180 Kb [1,2]. They are characterized by housing their genetic material in a capsid covered with a viral envelope and structurally defined by a cylindrical symmetry resembling a rod (hence its name baculo, which in Latin translates to rod). Typically, baculoviruses have a length of 200–400 nm and a width of 20–40 nm [1,2].

Viral entry into the insect host cells occurs thanks to the association of the glycoprotein GP64, an essential protein for viral infection, cholesterol and other membrane lipids and yet unidentified proteins [5]. GP64 is an envelope protein essential for cell-to-cell transmission of infection [6]. Deletion of gp64 gene results in viruses that cannot enter insect cells. Baculoviruses collect GP64 from the plasma membrane of infected cells as they bud during the infection cycle. This glycoprotein has been the main target for baculovirus display of foreign proteins and peptides [7]. However, other capsid proteins, such as VP39, have recently been used, successfully, for display [7]. VP39 is the major capsid protein, while GP64 concentrates in one pole of the rod (Figure 1) [8,9,10].

In a natural infection, baculoviruses adopt two different phenotypes. The first is as occluded virions, which are internalized into a crystal structure known as polyhedra, mainly consisting of the protein polyhedrin. The second viral phenotype is known as a budded virus. The budded virus is the main form of cell-to-cell infection, while occluded virions are released into the environments during insect lysis and may remain inside the polyhedra, retaining their infectivity for a very long time [11,12].

Baculovirus display technology consists of exposing a protein of interest on the surface of recombinant baculoviruses. Here, we describe the characteristics of baculovirus display technology and highlight its main advantages and drawbacks. Finally, we show evidence of the functional display of the nonapeptide bradykinin on the surface of recombinant baculoviruses, fusing the sequence coding for bradykinin to a second copy of gp64, driving the expression of the recombinant protein with the very strong polyhedrin promoter.

### 1.1. Characteristics of the Baculovirus Expression Vector System

The heterologous protein expression in insect cells using recombinant baculoviruses was first described in the 1980s [13]. Since then, the baculovirus expression vector system (BEVS) has been modified to improve its efficiency and diversify its applications. BEVS offers advantages over other expression systems, like incorporation of post-translational modifications like mammalian cells (e.g., phosphorylation and glycosylation), inherent safety (baculoviruses do not replicate in mammalian cells), high levels of protein expression, and scalability to mention a few [14]. The fundamental components of BEVS are an insect cell line and a recombinant baculovirus [15].

Most studied baculoviruses belong to the genera *Alphabaculovirus*, which comprises the lepidopteran Nucleopolyedroviruses (NPVs) [2]. Autographa californica multiple Nucleopolyedroviruses (AcMNPV) is the prototype baculovirus to produce recombinant proteins, and Bombyx mori NPV (BmNPV) is used to a lesser extent [16]. AcMNPV has an enveloped double-stranded circular DNA genome (134 kbp) packaged into a rod-shaped nucleocapsid. Baculoviruses have a biphasic life cycle, producing two virion phenotypes: the budded virus (BV) and occlusion-derived virus (ODV). BV is the one required for the virus replication in culture cells [17].

Sf9 and Sf21 insect cell lines, derived from pupal ovarian cells of the fall armyworm (Spodoptera frugiperda), are commonly used for BEVS. Also popular is the Tn5 cell line (High Five, Invitrogen), originated from ovarian cells of the cabbage looper (Trichoplusia ni), associated with higher recombinant protein expression, particularly of secreted proteins [18].

Recombinant baculovirus can be produced mainly by two strategies, homologous recombination and transposition. In the first case, insect cells are co-transfected with the viral DNA and a suitable transfer vector carrying the foreign gene. The second methodology benefits from the development of bacmids (bacterial artificial chromosomes), which are baculovirus genomes harboring a bacterial origin of replication, enabling the genome propagation in bacteria as plasmids [17].

A transfer vector is a plasmid that contains the gene of interest, usually under control of a strong viral promoter like polyhedrin (polh) or p10 [2]. These promoters are strong and direct the very late expression of non-essential genes for viral replication, allowing them to be replaced by foreign coding sequences during a transposition or recombination event [19]. Nonetheless, alternative promoters have been explored to improve protein expression yield/quality or to regulate temporal expression or cellular localization. Some examples are p6.9, vp39, and gp64 [20]. There are many commercially available transfer vectors, often only compatible with a particular baculoviral DNA or system, with different options of promoters.

Some commercially available systems harbor a double promoter in the same plasmid, as well as optimized features, such as useful purification tags and protease cleavage sites for subsequent tag removal. Some examples of transfer vectors are pBAC and pIEx (Novagen), pOET (Oxford Expression Technologies, Oxford, UK), and pFastBac and pDEST (ThermoFisher, Waltham, MA, USA); particularly, pTriEx plasmids (Sigma-Aldrich-Novagen, Saint Louis, MO, USA) offer triple host promoter for insect, bacterial and mammalian host cell gene expression. Examples of transfer vectors that use gp64 as promoter are pBAC-5 and pBAC-6, and there is one called pBACsurf-1 that targets gene insertion between gp64 signal peptide and coding sequences, thus favoring its surface display [19,21].

Over the years, accumulated knowledge of AcMNPV molecular biology has led to the development of many commercial baculoviral DNAs for homologous recombination. The original method utilized circular wild-type AcMNPV genome and was associated with a very low frequency of recombination (<1%), making it very difficult to identify and isolate the recombinant baculoviruses from the high background of parental viruses [16]. The insertion of a unique enzyme restriction site (Bsu36I) at the polh locus of AcMNPV DNA allowed its linearization. This change reduced wild-type virus infectivity and increased the recombinant baculoviruses efficiency to 30% [22]. However, several rounds of plaque assays were still required to purify them. The interruption of the essential open reading frame (ORF) 1629 favored the production of recombinant viruses to more than 90%; it was achieved by introducing point mutations to generate two additional sites for Bsu36I at locations that flank the polh locus in the upstream (ORF603) and downstream (ORF1629) genes. Triple digestion of this AcMNPV DNA variant induces partial deletion of ORF603 and ORF1629, and only recombination can reestablish virus replication [23]. This optimized baculovirus DNA was commercialized as BacPAK6 DNA (Takara-Bio, San Jose, CA, USA); importantly, the transfer vector must contain the sequence of foreign gene flanked with the sequence of the viral genome required to restore ORF1629. Many commercial systems are based on this principle, for example, BaculoGold (BD Biosciences), Bac-N-Blue (ThermoFisher, Waltham, MA, USA), BacMagic (Sigma-Aldrich-Novagen, Saint Louis, MO, USA), and flashBAC (Oxford Expression Technologies, OET, Oxford, UK) [15]. Some of these systems have additional baculoviral DNA variations, for example, introduction of enhancer elements (e.g., homologous regions hr), deletions of genes that can have an adverse effect on stability and secretion of recombinant proteins like chitinase (chiA) or cathepsin (v-cath), deletion of other non-essential genes such as p10, p26, and p74 to improve protein expression yield, insertion of anti-apoptotic genes or molecular chaperones to favor target protein stability, and folding and prolonging cell survival during infection to increase foreign protein expression [24].

The transposition strategy was pioneered by Luckow et al. [25], who developed a specialized strain of Escherichia coli (DH10BAC) containing a “baculovirus shuttle vector” or bacmid, which was transformed with a transfer vector harboring the gene of interest between attL and attR Tn7 restriction sites, and downstream the desired baculoviral promoter. This bacmid carries a mini-F replicon for single-copy replication in *E. coli*, a kanamycin resistance gene for selection, and a LacZ-mini-attTn7 transposon integration site [25]. In the presence of a helper plasmid that contains a Tn7-recombinase, site-specific transposition of the DNA sequences from the transfer vector to the bacmid results in gene insertion. The Bac-to-Bac baculovirus expression system (ThermoFisher, Waltham, MA, USA) is based on this strategy, that relies on transfer vectors known as pFastBac or pDEST. A recombinant bacmid is selected through antibiotic resistance and blue color screening in bacteria, purified by alkaline lysis, and analyzed by PCR before transfection into insect cells for baculovirus production [21].

It is important to mention that there are hybrid strategies, which combine optimized homologous recombination and bacmid technology (e.g., BacMagic and flashBAC systems). In this hybrid system, the AcMNPV genome lacks part of an essential gene (ORF1629) and contains a bacterial artificial chromosome (BAC) at the polh locus, replacing the polyhedrin coding region. BAC allows the viral DNA to be maintained and propagated within bacterial cells and the essential gene deletion prevents virus replication in insect cells unless there is successful recombination [16,26].

In general, using a commercially available system with vectors which include a second copy of the gp64 gene with a multiple cloning site between the sequence for the signal peptide and the rest of gp64, facilitates the rapid production of GP64 fusion proteins, such as pBACsurf-1. This system employs the gp64 promoter, ensuring the expression of the GP64 recombinant protein at a time of infection like the time when endogenous GP64 is expressed. Other systems using a strong (very late) promoter such as the polyhedrin promoter, results in the late production of recombinant GP64 which may result in less GP64 fusion protein present in the virion.

Using systems with a dual promoter facilitates the identification of the recombinant baculoviruses, since a fluorescent protein can be included in the second promoter for expression and identification of fluorescent cells.

### 1.2. Baculovirus Display Strategies

Baculovirus display technology is a novel method for displaying peptides and proteins on the surface of the baculovirus. It is an effective method for creating and displaying epitope-specific antibody binders or any other peptide or protein of interest for use in immunoassays, vaccines, and other biotechnological applications. The baculovirus display technology has been widely used in research and development of antibodies, vaccines, and biopharmaceuticals [7,27].

Different strategies have been used for the display of several proteins on baculovirus, allowing presentation of proteins of interest on the capsid or viral envelope.

The first baculovirus expression vector system (BEVS) developed was designed on the *Autographa californica* multiple nucleopolyhedrovirus (AcMNPV), and the insertion of heterologous proteins have been carried out as a fusion on the major envelope glycoprotein GP64. The different strategies have used the fusion of the foreign proteins to the amino terminal of GP64, just after the signal peptide, and most recently, in the interest of seeking better strategies, a number of new combinations, such as the use of only the signal peptide and the transmembrane domain, or the use of signal peptides with high secretion levels as the honeybee melittin signal peptide have been proven successfully [28].

Another strategy has been the fusion to capsid proteins of the baculovirus, such as VP39, the most abundant nucleocapsid protein of AcMNPV. Fusion at the amino or the carboxy terminus of VP39 has resulted in the successful display of the protein of interest [7].

The use of heterologous membrane anchor proteins as VSV-G (vesicular stomatitis virus glycoprotein), alone or as carrier of fusion peptides and proteins has also been carried out. The measles virus receptor and the hemagglutinin (HA) and neuraminidase proteins from influenza virus have been displayed using this method [28,29].

### 1.3. GP64 Glycoprotein

GP64 is a trimeric class III viral fusion glycoprotein found in the envelope of many members of the family Baculoviridae. The protein has a molecular weight of 64 kDa and is essential for cell-to-cell propagation of the budded virus and for systemic infection in host insect [6,30]. Deletion of gp64 results in replication-deficient baculoviruses.

Considering that GP64 is a transmembrane protein that exposes an external domain, it can be used to display a protein of interest in the budded baculovirus (BV) envelope. It is possible to design a recombinant GP64 fused to the protein of interest or short-length peptides, incorporating the recombinant protein into the BV structure after infection of insect cells [31].

Other strategies include using the transmembrane™ region and cytoplasmic tail domain (CTD) of GP64 instead of generating fusion proteins for full-length GP64. Using only these regions has been shown to be sufficient for the insertion of the protein of interest in the plasma membrane of infected insect cell, as well as on the BV. The goal of this strategy is to allow for smaller transfer vectors and faster cloning procedures, as well as to increase the number of molecules deployed [32].

### 1.4. Surface Display by Fusion to Complete GP64

The first strategies for protein deployment in baculoviruses were constructed from the fusion of the protein of interest with full-length GP64 [33,34,35,36]. In these studies, it was shown that the fusion of the proteins of interest to GP64 does not affect the maturation of the virus or the localization of GP64 in the plasmatic membrane of infected cells. Furthermore, BV carried the recombinant protein on its surface, demonstrating functional recombinant protein display [37,38].

Several studies evaluated the best alternatives in which the proteins should be fused to GP64 [38,39,40]. These studies demonstrated that fusing the protein of interest to the full-length GP64 is more efficient for the deployment of the fusion protein on the surface of the baculovirus [33].

In addition, the use of full-length GP64 allows the display in baculoviruses of relatively large proteins such as VP1, the major viral capsid protein of foot-and-mouth disease virus (FMDV), a 74 kDa protein [34]. Other examples include the glycoprotein D of Bovine Herpesvirus 1 (BoHV-1) [35] and Vp1 of Enterovirus 71 (E71) [36]. These examples demonstrate the displaying of proteins to generate an immune response and the production of antibodies. Several examples show the conservation of the biological function of several proteins when expressed fused to the full-length GP64, such as avidin, which provides a binding site to coat the virus with desired biotinylated ligands [41].

The advantages of using this method is stressed in the fact that all the sequences necessary for the transport and maturation of proteins are present in the complete sequence of gp64, therefore the proteins of interest fused to GP64 are directed to the plasmatic membrane and to the viral envelope [31].

However, the use of full-length GP64 may generate some problems in the cloning process due to the length of the transfer vector [31], for this reason, it has been proposed the use of specific regions of GP64 for the construction of recombinant baculoviruses.

More recently, it has been shown in several research papers that the use of GP64 regions such as the signal peptide (SP), the transmembrane region (TM), or the cytoplasmic tail domain (CTD) are sufficient for successful display of the recombinant protein on the baculovirus envelope of budded viruses.

The objective of this strategy is to reduce the size of the transfer vectors to allow faster and more efficient cloning and to improve recombinant protein surface display [32]. It has been shown that using recombinant GP64 proteins harboring only some domains from GP64 result in reduced surface display of the protein of interest due to interference in trimer formation between the recombinant protein and wild type GP64 [42].

Several strategies have been explored to overcome problems with recombinant GP64 display when using fragments of this protein [43]. In a recent study, displaying the glycoprotein E of varicella-zoster virus (VZV; also known as human herpesvirus 3) that causes chickenpox, a polyhistidine (6 × His) tag at the N-terminus of GP64 was inserted and combined with the signal peptide (SP) of this protein to enhance surface display [43].

Another example of use of specific regions of GP64 include the fusion of the Zika virus envelope glycoprotein (gE) with the signal peptide (SP) and transmembrane region of GP64 [39] or the hemagglutinin (HA) of the H5N2 avian influenza virus using its own TM and fused to the SP and CTD of GP64 [39].

In most of the studies on baculovirus display, the production of recombinant proteins was carried out in the presence of wild-type GP64. However, the presence of GP64 in budded baculoviruses is not entirely desirable, since the protein of interest competes with wild-type GP64 expressed on the baculovirus envelope leading to low efficiency in the presentation of the target protein on the viral surface [44].

In previous studies it was observed that in the absence of GP64, the viruses are incapable of budding from the cell and therefore of spreading the infection cell to cell, in this sense the virions are produced in small quantities and are not infectious [45]. Therefore, GP64 is critical for both production and entry of virions. Interestingly, the essential functions of GP64 may be replaced using heterologous proteins that replace GP64 function for viral entry. VSV-G is a viral protein that promotes virus entry into cells in a GP64-null baculovirus [46].

### 1.5. Using Peptide Insertions on GP64

As demonstrated by Ernst et al. [37], it is possible to insert small peptides (up to 12 amino acids) directly on native gp64 coding sequences without interfering with surface localization of the modified GP64 and conserving recombinant virus infectivity. Inserted peptides were the HIV-1-GP41 epitope sequence (ELDKWA). This approach supports the potential use of recombinant baculoviruses for direct antigen display purposes for the immunization of animals to produce antibodies, or in diagnostic or research fields. In addition, display of affinity tags is a potential method to purify recombinant baculovirus or, in the case of co-expression with streptavidin fusion proteins, target them to the cell surface by streptag-streptavidin interactions. In a following study, Spenger et al. [38] tested 17 different positions in GP64 to insert the epitope ELDKWA, obtaining successful replication for 13 recombinant viruses. Probably in the negative cases, peptide insertion interferes with GP64 folding or oligomerization, and thus these proteins may not reach the cell surface, ultimately compromising virus budding and infectivity.

In another study, it was possible to display a peptide of 23 amino acids into GP64. This peptide is from VP1 protein of foot-and-mouth disease virus (FMDV) and harbors an RGD-motif, which is important for internalization of many viruses in mammalian cells. Therefore, its display was expected to enhance baculovirus transduction. However, overall GP64 expression decreased in recombinant virus, and in consequence viral uptake by mammalian cells [47].

These studies suggest that insertion position in gp64 as well as peptide size are critical parameters to consider for the correct display of proteins of interest. An advantage of peptides directly engineered into native gp64 is that could have a higher surface expression, compared with fusion peptides to N-terminal region of a second copy of GP64 [32].

### 1.6. Capsid Display by Fusion to VP39

Baculovirus display technology has not been limited to the display of proteins in the viral envelope (GP64 fusions), since the capsid components have been used for the display of several proteins. Particularly the VP39 protein, the major component in the baculovirus capsid, represents an excellent target for protein display.

Kukkonen, et al. (2003) demonstrated that GFP fused to either the N-terminus or C-terminus of VP39 resulted in functional recombinant baculoviruses [7]. Laakkonen, et al. (2009) have fused mCherry protein to the C-terminus of the capsid structural protein P24, to follow the internalization process of baculoviruses into mammalian cells [48]. Chen, et al. (2011) have improved transduction processes of the baculovirus by fusing VP39 to the protein transduction domain (PTD) of HIV TAT protein and thereby forming a nuclear membrane penetrating baculovirus [49].

VP39 has been proposed for different biotechnological applications in baculovirus display. Song, et al. (2010) have worked on the development of recombinant baculoviruses that display genetically modified polypeptides that bind to organic compounds, better known as GEPI. In this study, VP39 was fused to a chelating peptide that recognizes Zinc oxide, which could favor the nucleation and assembly of hybrid nanomaterials based on ZnO nanoparticles [50].

VP39 has also been shown with great potential to display foreign proteins in baculoviruses, leading to vaccine development. A recombinant baculovirus expressing an ovalbumin fragment fused to the VP39 capsid protein has been shown to act as an adjuvant and antigen vector by promoting CD4 and cytotoxic T cell responses against OVA, as well as inducing cell maturation of dendritic cells and production of inflammatory cytokines in mice immunized with VP39-ovalbumin. This response has been sufficient to enhance the survival of mice carrying ovalbumin-expressing melanoma cells [51,52].

### 1.7. Baculovirus Display with Complete Proteins

The display of foreign proteins on baculoviruses can be achieved without the need to fuse with GP64 or VP39, it has been possible to display full-length proteins that contain their own signal peptide and their own transmembrane domain, displaying native protein in the baculovirus envelope for use in different applications.

The possibility of using the native protein has some advantages, the first of which is that it makes the construction of cloning vectors faster, using the properties of the protein for functional display on the baculovirus envelope. On the other hand, the misfolding of fusion proteins is more likely to present problems, which is solved by using native foreign proteins instead of fusion proteins.

The application of this strategy has been used for the design of veterinary vaccines, the display of full-length VP28 on the baculovirus surface was achieved. VP28 is one of the major envelope proteins of the white spot syndrome virus (WSSV) and plays a crucial role in the viral infection of a wide range of hosts including freshwater shrimp, lobsters, freshwater crabs, and several species of marine crabs [53]. The immunization of shrimp by these baculoviruses that display VP28 on their envelope resulted in a survival of 86.3% of shrimp infected with WSSV, which represents a great advance considering that infection of shrimp by this virus can have up to 100% mortality [53]

Other examples of full-length foreign protein display include the generation of a recombinant baculovirus that displays full-length VSV-G [54], *Toxoplasma gondii* roptria protein 4 (ROP4) [55] and the hemagglutinin (HA) from the H5N2 avian influenza virus [56].

However, although the display of complete and unmodified proteins in the baculovirus envelope is possible, it seems to be an inefficient strategy in terms of the number of copies of the protein of interest incorporated into the viral envelope, this was demonstrated by comparing two recombinant baculovirus displaying the HA of the H5N2 virus, both baculoviruses displayed the full length HA with their own TM, the only change was the replacing the HA CTD region with GP64 CTD region. This resulted in higher HA expression in baculoviruses using the CTD region of GP64 [57].

### 1.8. Advantages and Applications of Baculoviral Surface-Display Technology

One of the main advantages of using baculovirus display is the fact that recombinant baculoviruses are produced using the well-characterized and extensively used baculovirus-insect cell expression system (BEVS) [15]. This system is easy to use, inexpensive and yields large amounts of recombinant protein, while preserving the posttranslational modifications such as glycosylation, phosphorylation, and others [15]. Even though insect cells do not use the same carbohydrates as mammalian cells in protein glycosylation, most of the recombinant proteins produced by BEVS are functional [15]. There is a large list of commercially available enzymes, hormones, and vaccines produced with BEVS.

Another advantage is that baculoviruses are large viruses, and therefore are easier to purify by ultracentrifugation compared to smaller viruses such as adeno-associated virus (AAV) and other viruses also used to display proteins of interest [58,59].

Compared to other eukaryotic systems, the baculovirus expression system produces higher- expression levels and lower costs compared, for instance, to mammalian cells. Insect cells are easier to grow in culture or bioreactors and cell culture media is less complex and inexpensive compared to media used on other eucaryotic cells.

Finally, because baculoviruses are insect replicating viruses and are unable to replicate in mammalian cells, they have a high level of biosecurity compared to viruses that naturally infect mammalian cells, such as adenovirus, lentivirus, and others [28].

### 1.9. Baculovirus as Antigen Carriers

The ability to introduce large fragments of heterologous DNA into the baculovirus genome, together with the use of chimeric proteins on their surface, make baculoviruses an excellent tool as antigen carriers.

The process in which the natural envelope proteins of the virus are replaced with surface proteins of another virus is known as pseudotyping. This tool has been widely explored in baculoviruses thanks to its high versatility and ability to display foreign proteins in its surface. The most widely used baculovirus pseudotyping method is based on the use of the vesicular stomatitis virus glycoprotein (VSV-G). Several investigations have shown that baculoviruses that carry the VSV-G protein together with the co-expression of GP64 improve their transduction capacity in vertebrate cells [31].

Baculoviruses displaying VSV-G under the polyhedrin promoter enter PK-15 cells more efficiently, which improves gene delivery. Examples include the delivery and expression of the porcine circovirus type 2 (PCV2) capsid protein, resulting in the robust generation of antibodies against PCV2 in immunized animals [60]. In more recent work involving the use of pseudotyped baculoviruses with VSV-G to improve gene delivery in mammalian cells, displaying full-length VSV-G in the baculovirus envelope driving the expression of H5N1 hemagglutinin (HA) under the CMV promoter, results in efficient gene delivery and its consequent expression and generation of neutralizing antibodies [20].

The application of baculoviruses as carriers of antigens and the use of novel strategies to display proteins on their envelope have allowed to improve the specificity of therapies directed at the treatment of cancer. One of the first works that deals with baculoviruses as a vector for the development of cancer therapies is based on the construction of a baculovirus that display the sequence of the LyP-1 tumor localizing peptide and the reading frames of the transmembrane (TM) and cytoplasmic (CT) domains from VSV-G. LyP-1 specifically accumulates in lymphatic vessels associated with tumor cells, and is internalized by MDA-MB-435 human carcinoma cells and tumor lymphatic endothelial cells. Therefore, baculoviruses carrying LyP-1 can improve tropism for cancer cells and could be a good alternative for gene delivery in tumors [25].

The use of baculoviruses to carry proteins that are specifically recognized by a select group of cells has been shown to allow the development of effective and selective therapies. A baculovirus was designed to display anti-HER2 single-chain variable domain fragment (scFv) and the gene that expresses apoptonin under the CMV promoter. The expression of anti-HER2 allows the selective binding of baculovirus to tumor cells, since HER2 is a transmembrane receptor tyrosine kinase that is overexpressed in 30% of all cases of breast cancer and aggressive tumors. Once the baculovirus specifically enters cancer cells, apoptonin expression begins leading to tumor cell death [61].

It is also possible to mediate the transduction of cells of the immune system such as dendritic cells (DC), which was achieved through the generation of a baculovirus that displays the Fc region of human IgG in its envelope [62]. Cells of the immune system such as DCs express receptors for Fc (FcR). These receptors are a family of membrane proteins that bind to the constant region of immunoglobulins (Ig) in the region of the crystallizable fragment (Fc) and mediate phagocytosis and antigen presentation. These examples of protein display by recombinant baculovirus and the driving of foreign gene expression in selective cells or tissues clearly highlight the potential of recombinant baculovirus for the display and for gene therapy [27,28].

### 1.10. Development of Vaccines Based on Baculovirus Display

Baculoviruses have been widely used to produce vaccines as expression systems combined with insect cell cultures [24]. There are three main methodologies for vaccine development using baculoviruses. The first one is the generation of subunit vaccines, which consist of using the baculovirus as an expression system in insect cell cultures [63,64]. The objective of this method is to express the protein taking advantage of the high expression levels attained with baculovirus-insect cell systems. This system has shown the advantages of preserving the natural post-translational modifications, such as glycosylation. This method has been used to produce several commercially available vaccines [64,65].

The second method employed for vaccine production is the display of antigens on the surface of baculoviruses which are used as immunogens (Table 1). This system consists of using baculoviruses to carry the antigens of interest on their surface for immunization using purified recombinant baculoviruses [32,40]. Antigen display on the surface of recombinant baculovirus has 2 different modalities: a) membrane display using some baculovirus protein as a base and b) membrane display with transmembrane regions of exogenous proteins (Figure 1) [32,66]. Some of the major advantages of these methods are the conservation of protein folding and the fact that baculoviruses are naturally incompetent to infect mammalian cells [28,52,67]. Different immunization routes have been explored, including intramuscular, nasal, oral, and subcutaneous applications [36,68,69].

The third method for using baculoviruses as vaccine carriers uses pseudotyped baculoviruses (Figure 1). This method combines the two systems described above. It consists of displaying proteins in the baculovirus membrane (mainly proteins that help baculovirus entry into target cells) in combination with expression cassettes to induce the synthesis of antigens of interest in the target cell [63,70]. In addition to the advantages described above, the inclusion of tissue-specific promoters induces the expression of the antigen of interest in selected cells [27,70].

### 1.11. Display of Bioactives Peptides

There are many examples of baculovirus display of foreign proteins. Even small peptides such as the hexahistidine (His6) tagged ZnO binding peptide fused to VP39 has been shown recently [50].

However, after an extensive search of the literature, we did not find any examples of the successful display of small bioactive peptides. Displaying a small peptide as a fusion protein to the large GP64 or any envelope baculovirus protein represents a huge challenge. Most likely the small peptide will be hidden inside the complex structure of the baculovirus glycoprotein used to display it.

Most surprisingly, we found functional display of the nonapeptide bradykinin fused to GP64. Figure 2A shows the expression cassette we used for the display. The cDNA sequence coding for bradykinin nonapeptide was inserted between the signal peptide (SP) and the full-length sequence of gp64 (Figure 2A).

Bradykinin (Bk) is a 9 amino acid peptide that promotes vasodilation via the release of nitric oxide and prostacyclin [103]. Bk actions occur via the binding to either bradykinin type 1 or type 2 receptors [103]. Both receptors are the typical cell surface, G-protein coupled receptors of the seven-transmembrane domain family [103]. Association between bradykinin and the B2 receptor results in the rapid increment of intracellular calcium driven by the production of IP3 after phospholipase C activation [104].

To study the efficiency of displaying bradykinin on the surface of recombinant baculovirus, we first expressed the B2 human receptor on the surface of insect cells using a recombinant baculovirus carrying this gene [105]. B2 couples to the endogenous insect cell G-protein and induces the mobilization of intracellular calcium upon receptor activation [105]. In this manner, we can use intracellular calcium increments as a reporter of functional association between the baculovirus displaying bradykinin and the B2 receptor previously expressed in insect cells using a different baculovirus carrying the cDNA for this human receptor [105]. First, we demonstrated functional expression of the B2 receptor using commercially available pure bradykinin to activate it. Upon bradykinin addition, intracellular calcium rises in insect cells expressing the human B2 receptor, but calcium increments were not detected in cells that did not express B2 (Figure 2B). As a control we used a baculovirus that displayed the green fluorescent protein (GFP) fused to GP64 (Figure 2B, Bac-GFP). The addition of Bac-GFP did not induce any detectable increments in intracellular calcium of cells expressing the human B2 receptor. Similar results were obtained when using only saline solution without any baculovirus present (Figure 2B, PBS).

Most interestingly, addition of purified baculovirus displaying bradykinin (Bac-Bk) induced a slow rise in intracellular calcium, when compared to the fast rise induced by pure commercially available bradykinin (Figure 2C). To further demonstrate that the rise in calcium was due to the activation of the B2 receptor, we used the potent bradykinin antagonist HOE-140 [106]. Application of HOE-140 prior to bradykinin stimulation prevented the increment in intracellular calcium (Figure 2D). A similar result was obtained when using HOE-140 prior to the addition of the Bac-Bk recombinant baculovirus (Figure 2E).

A robust dose-response curve was obtained using increasing concentrations of commercially available bradykinin (Figure 2F). Using sequential dilutions of the Bac-Bk virus we could find a similar dose-response as that obtained with bradykinin (Figure 2G).

These results show for the first time the successful display on the surface of recombinant baculovirus of a small (nine amino acids long) bioactive peptide, opening the possibility of producing different bioactive peptides and proteins using baculovirus display in combination with the baculovirus/insect cell expression system. In this example we utilized GP64 to display bradykinin, however, other viral proteins can be explored for this purpose, such as VP39 and other proteins from the viral capsid. However, because VP39 and other capsid proteins are located internally in the baculovirus they may not function properly for the display of peptides or proteins that must be externally displayed to bind its specific receptor. This was the main reason for using GP64 for the display of bradykinin, something that needs to be considered in the display strategy.

## 2. Discussion and Conclusions

Baculoviruses are insect-specific viruses that have been used extensively in research, the production of biopharmaceuticals, and biotechnological applications. They are a group of large, double-stranded DNA viruses that infect the larvae of insects, such as moths and butterflies. Baculoviruses have been developed as insect cell expression systems to produce recombinant proteins due to their ability to infect and replicate within insect cells with high efficiency, resulting in the production of large amounts of proteins of interest. Insect cell expression systems using recombinant baculoviruses have been used to produce a wide range of recombinant proteins with various functions, including vaccines which are commercially available. The baculovirus expression system is advantageous compared to other expression systems because it is relatively inexpensive and easy to use, while also providing high yields of recombinant proteins with functional activity due to correct post-translational modifications. Additionally, the baculoviral vector can express foreign genes in vivo in eukaryotic cells, opening the possibility of using recombinant baculoviruses for gene therapy or gene targeting.

Baculovirus display technology allows the presentation of functional peptides and proteins in the viral envelope or capsid. The GP64 and VP39 proteins have been the two main targets for the baculovirus display system. However, there is a wide repertoire of baculovirus structural proteins to be explored as a display system, including P24 and many others.

In this paper, we review the different strategies for protein deployment in baculoviruses and their biotechnological uses, and we also report the functional display of the small peptide bradykinin fused to GP64. This novel result opens the possibility of displaying other small bioactive peptides, such as the atrial natriuretic peptide (ANP). ANP is a 28-amino acid peptide involved in sodium exclusion in kidney and cardiac function [107]. Another candidate is the nonapeptide oxytocin, a hormone with multiple functions including social bonding, reproduction, and childbirth [108]. Recent studies have shown that oxytocin treatment may play a role in improving autism [109] and other mental disorders such as post-traumatic stress disorder (PTSD) [110]. Relatively larger peptides such as leptin may be also relevant targets of baculovirus display. Leptin is a hormone predominantly made by adipose cells which has gained attention in recent years because transgenic mice unable to produce leptin show obesity, constant hunger, and lethargy [111].

The future of baculovirus display is bright with great opportunities in the research of vaccine design, gene therapy, peptide therapy, diagnostics among many others.

## Figures and Tables

**Figure 1 viruses-15-00411-f001:**
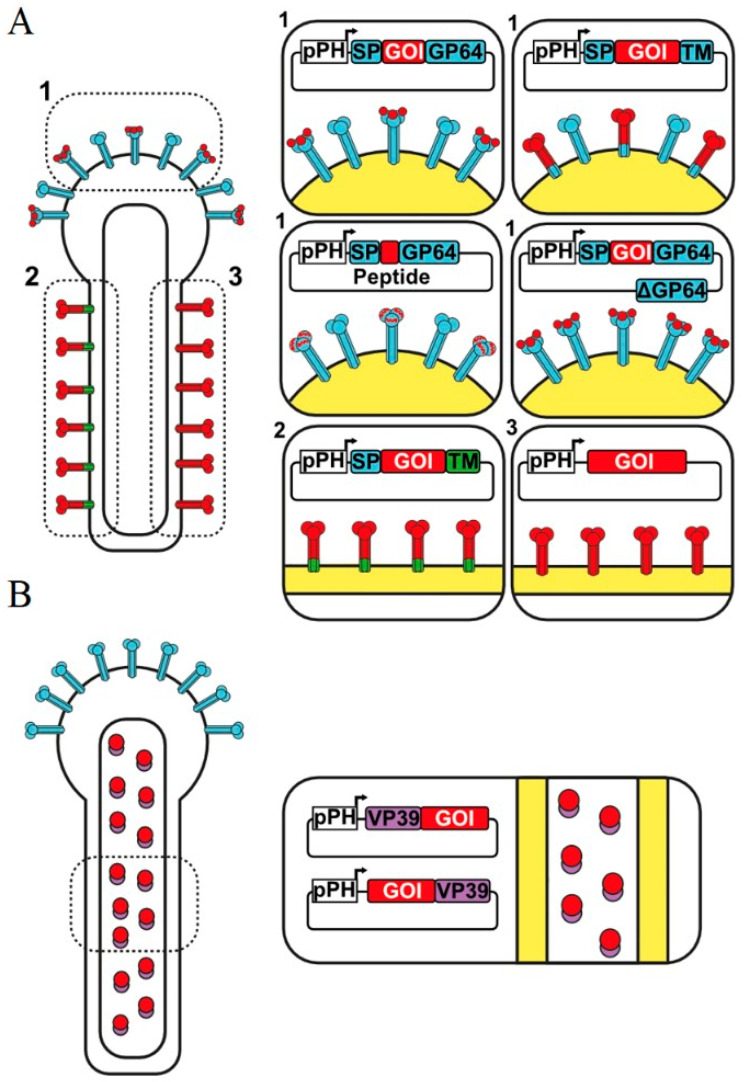
**Protein display strategies in the “Baculovirus display” system.** (**A**) Envelope display of full-length proteins, domains, or peptides fused to the full length GP64 protein or fragments as the signal peptide (SP) and transmembrane domain (TM) (1). Baculoviruses in which GP64 has been deleted (ΔGP64) have been shown to be functional when GP64 is expressed as a recombinant protein fused to a gene of interest (GOI). (2 and 3) Other strategies consist in display proteins associated with transmembrane domains of non-baculovirus proteins, or those that have their own transmembrane domain. (**B**) Display of capsid proteins, fused to the amino or carboxyl terminus of the VP39 capsid protein.

**Figure 2 viruses-15-00411-f002:**
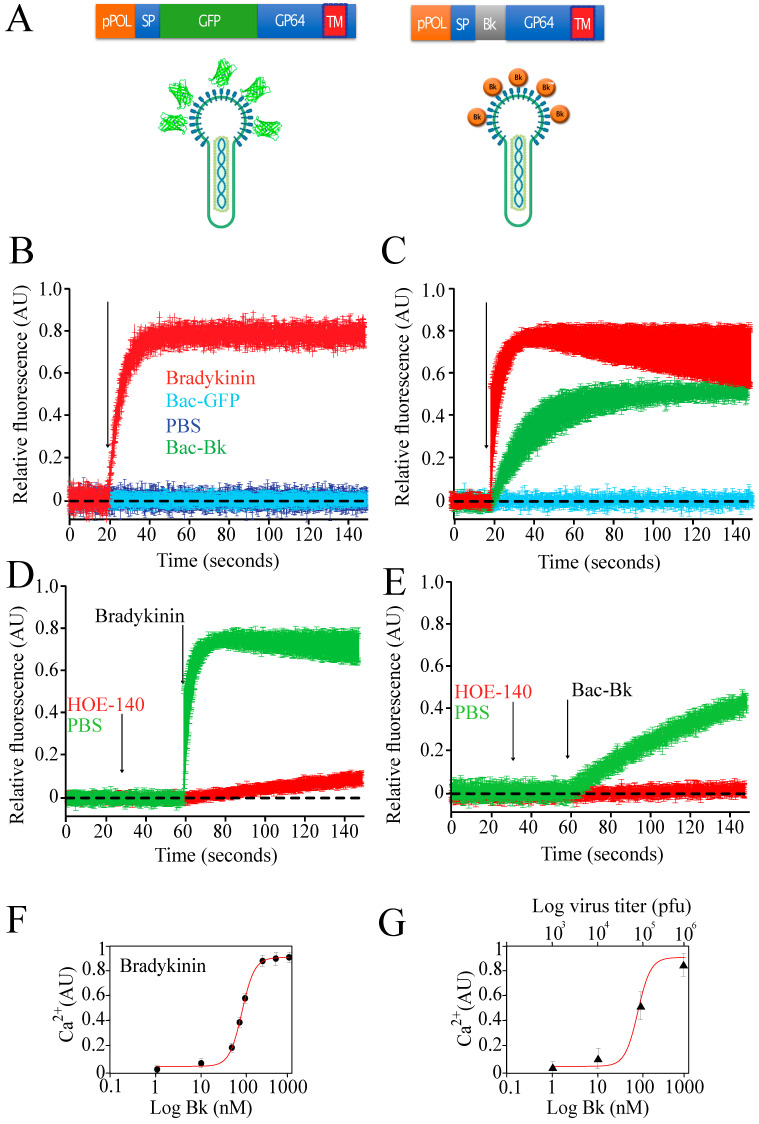
**Functional display of the nonapeptide bradykinini on the surface of recombinant baculoviruses.** (**A**) cartoon displaying the constructs for the expression of a second copy of recombinant GP64 (Bac-GFP and Bac-Bk) under the polyhedrin promoter. (**B**) Sf9 insect cells expressing the human type 2 bradykinin (B2) receptor (produced with a recombinant baculovirus carrying the cDNA for human B2) were stimulated by the addition of commercially available bradykinin (Bk, Sigma). FURA-2 loaded cells responded to Bk with a rapid elevation in intracellular calcium (red line). (**C**) insect cells expressing B2 were exposed to the Bac-Bk virus displaying the nonapeptide bradykinin on its surface fused to GP64. Cells responded with a slower elevation in intracellular calcium (green line). Cells stimulated with the recombinant baculovirus expressing GFP fused to GP64 did not respond to the stimulation, indicating that the response to Bac-Bk was due to bradykinin (blue line). (**D**) bradykinin stimulation of cells expressing the human B2 receptor was completely blocked with the potent antagonist HOE-140 (red line). (**E**) similarly, the calcium response induced by Bac-Bk was blocked by the same antagonist (red line). All data presented are the mean +- standard deviation obtained from at least 10 independent measurements from 5 different infections. (**F**) typical dose-response curve to bradykinin in cells expressing the human B2 receptor. (**G**) a dose-response curve was also obtained using different dilutions of the Bac-Bk recombinant virus. Data points represent the mean +- standard deviation obtained from at least 10 independent measurements from 5 different infections. For these experiments recombinant baculoviruses were purified by ultracentrifugation.

**Table 1 viruses-15-00411-t001:** Experimental vaccines based on baculovirus display.

Platform Design	Displayed Antigen	Disease	Model/Dose	Adjuvant	Route of Administration	Immune Response	Reference
Virus vaccines							
Display HA protein cloned under ie1 promoter from WSSV	HA	Influenza H5N1	Mouse/3 doses (1 × 10^8^ pfu) inactivated or live recombinant baculovirus	With or without rCTB Reverse micelle (liposomes of phosphatidylcholine) based carried vehicle	Oral	Specific serum IgG and mucosal IgA antibodies. High HI titers. Cross neutralization. Protection (100% for live baculovirus) in challenge study	[71,72]
Display HA protein cloned under ie1 promoter from WSSV	HA	Influenza H5N1	Mouse/2 doses (5.6 × 10^6^ pfu)	With or without rCTB	Oral	Production IgG, IgG1, IgG2a and IgA systemic; IgG and IgA at mucosal sites. Induces germinal center B cell response. Significant neutralizing antibody titers. Partial protection against lethal challenge.	[73]
Display combinations of full-length or ED of HA fused to SP or TM or CTD domains of GP64 derived from AcMNPV, all under Pph	HA	Influenza H1N1 and H5N1	Mouse/2 doses (1 × 10^7^ or 1 × 10^8^ or 2 × 10^7^ or 4 × 10^6^ pfu)	Baculovirus act as adjuvant	i.m.	Elicits HI activity. High levels of IFN-γ secreting and HA-specific CD8+ T cells. Protection around 60–100% in challenge study	[74]
Display HA cloned between SP and CTD of GP64 under Pp10	ED and TM domains of HA	Influenza H5N2	Mouse/3 doses of 200 μL (1 × 10^8^–1 × 10^10^ pfu)	Baculovirus act as adjuvant	i.m., s.c., and i.n.	High specific IgG2a and IgG1 neutralizing antibodies. Induces high HI titers. Induces IgA antibodies. Elicits cellular immune response (Th1 and Th2).	[69]
Display HA protein under ie1 promoter from WSSV	HA	Influenza H7N7 or H6N1	Mouse/H7N7: 2 doses (2^8^ HA units). H6N1: 2 doses (1 × 10^9^ pfu/mL)	i.n.: baculovirus act as adjuvant s.c.: Montanide ISA 201 VG	i.n. and s.c.	High levels of systemic IgG and mucosal IgA. Enhances neutralizing activity. High levels of IFN-γ and IL-4. Protection (100%) in challenge study.	[75,76]
Display VP1 protein fused between C-terminus of GP64 from AcMNPV and the N-terminus of TM region from H3N2 under ie1 or Pph promoters	VP1 from HEV71	HFMD	Mouse/2–3 doses (1 × 10^7^ − 1 × 10^8^ pfu)	Bilosomes (lipid-based vesicles incorporating bile salts) or Freund’s adjuvant	Oral and s.c.	Specific systemic IgG and mucosal IgA immune responses. High neutralizing antibody titers.	[36,68]
Display VP1 protein cloned into the C-terminus of NA from influenza A under ie1 promoter	VP1 from HEV71	HFMD	Mouse/3 doses (1 × 10^8^ pfu)	Freund’s adjuvant or Montanide adjuvant	s.c.	Specific IgG antibodies. High neutralizing antibody titers. Produces significant levels of IFN-γ. Protection (100%) against lethal challenge.	[77]
Display VP2 protein fused to TM and CTD of GP64GP64 protein under Pp10	VP2 protein from CPV-2	Parvoviruses	Mouse/2 doses (3 × 10^8^ pfu)	Baculovirus act as adjuvant	i.p.	Elicits immune response. High levels of virus neutralization.	[78]
Display VP28 protein fused to SP and C-terminus TM of GP64 protein under ie1 promoter	VP28	WSSV	Shrimp/ Injection: 2 doses (50 μL 1 × 10^8^ pfu/mL) Oral: pellet feed continuously for 7 days (3 mL 1 × 10^8^ pfu/mL). Immersion: seawater 2 L (3 mL with 1 × 10^8^ pfu/mL)		Injection, oral and immersion	Injection: 73–86% protection post challenge Oral: 76–82% protection post challenge Immersion: 68–75% protection post challenge	[53,79]
Display nine truncated CyHV-2 membrane glycoproteins, fused between SP and TM of GP64 under chicken β-actin promoter.	Membrane glycoproteins ORF25, ORF25C, ORF25D, ORF30, ORF124, ORF131, ORF136, ORF142A and ORF146	Herpesviral haematopoie-tic necrosis disease	Carp/30 mL of 6 × 10^5^ TCID_50_/mL were diluted into 5 L freshwater		immersion	Expression of IL-11, INF-α, INF-γ and a complement component gene C3 significant up-regulated. Protection greater than 70% in challenge study.	[80]
Display RVG protein cloned one copy under Pph and another copy under PCMV.	RVG	Rabies	Mouse/2 doses (1 × 10^8^ IFU)	Baculovirus act as adjuvant	i.m.	High levels of virus-neutralizing antibodies. Elicits cellular immune response Protection (100%) against rabies viral challenge.	[81]
Display ORF2 protein cloned between SP and TM domains of GP64 under Pph and PCMV promoters and, VSV-G under Pp10.	Cap or ORF2 from PCV2	PMWS	Mouse/2 doses (1 × 10^9^ pfu/mL)	Baculovirus act as adjuvant	i.m.	IgG antibodies. High specific neutralizing antibody titers. Production of IFN-γ.	[82]
Display P97R1P46P24 protein under Pph promoter; Cap protein cloned under Pph promoter. Genes were cloned between SP and TM of GP64.	P97R1, P46 and P24 from Mhp; Cap from PCV2	MPS and PMWS	Mouse and swine/3 doses for mice (1 × 10^8^ pfu/mL) 2 doses for piglets (1 × 10^9^ pfu/mL)	Baculovirus act as adjuvant	s.c.	High levels of P97R1P46P24-Cap specific IgG. Induction of cellular immune response.	[83]
Display GP5 protein under Pp10 and Cap protein under Pph. Both proteins cloned between TM and CTD domains of GP64.	GP5 from PRRSV and Cap from PCV2	PRRS and PMWS	Mouse and swine/2 doses for mice and piglets (1 × 10^9^ pfu)	Baculovirus act as adjuvant	i.m.	High GP5 and Cap antibody titers. High virus neutralization titers. Induces specific cell-mediated immune response.	[84]
Display ORF2a, ORF3, ORF4 and ORF5 proteins between C-terminus SP of GP64 and N-terminus TM HA from H3N2 that continuous of CTD of GP64, under ie1 promoter	ORF2a (36-207aa), ORF3 (27-265 aa), ORF4 (23-157aa) and ORF5 (33-200aa or without 64-90/108-129 regions)	PRRS	Mouse/2 doses of 100 μL (1 × 10^8^ pfu)	Baculovirus act as adjuvant	s.c.	Production of high specific and neutralizing antibodies.	[85]
Display S protein or S1 subunit cloned between SP and TM of GP64 protein, under Pp10.	S and S1 from PEDV	PED	Mouse and swine/ mice: 2 doses of 200 μL (1 × 10^9^ TCID_50_/mL) Piglets: 2 doses of 2 mL (1 × 10^9^ TCID_50_/mL)	Baculovirus act as adjuvant	i.m.	Systemic S-specific IgG. S-specific neutralizing antibodies. Decrease severity of illness in challenge study.	[86]
Display E glycoprotein linked to SP and TM domains of GP64 under Pp10	E from JEV	JE	Mouse and swine/2 doses for mice and piglets (1 × 10^9^ pfu)	Baculovirus act as adjuvant	i.p. and i.m.	Produces E-specific antibodies. Induces neutralizing antibody response and protective immunity toward a lethal challenge. Induces cell-mediated immune response.	[87]
Display E2 glycoprotein fused to TM and CTD domains of GP64 under Pp10.	E2 from CSFV	CSF	Mouse/3 doses (1.5 × 10^8^ pfu)	Freund’s adjuvant	i.p.	High E2 antibody titers. Significant level of neutralizing antibodies.	[88,89]
Display NS3 protein fused to TM and CTD domains of GP64 under Pp10.	NS3 from CSFV	CSF	Mouse/3 doses (1.5 × 10^8^ pfu)	Freund’s adjuvant	i.p.	High specific NS3 antibody titers.	[90]
Display E protein fused to SP and TM domains of GP64	E	ZIKV	Mouse/3 doses (15 μg)	Freund’s adjuvant	i.p.	High levels of systemic IgG. High neutralizing antibody titers. High levels of IFN-γ and IL-4. Protection greater than 80% in challenge study.	[39]
Display S1 peptide between SP and TM of GP64 under Pph and PCMV promoters and, VSV-G under Pp10.	ED of S1 glycoprotein from IBV	IB	Chicken/2 doses of 200 μL (2 × 10^8^ pfu)	Baculovirus act as adjuvant	i.m.	Strong humoral and cell-mediated immune responses. Induces cytotoxic T lymphocyte responses. Protection greater than 80% in challenge study.	[91]
Display D glycoprotein fused to GP64	ED of gD from BHV-1	BHV-1	Mouse/3 doses of 200 μL (5 × 10^8^ pfu)	Freund’s adjuvant	i.p.	High specific IgG and neutralizing antibodies.	[35]
Display S protein fused between SP of GP64 and TM domain of VSV-G under Pph	ED of S protein from SARS-CoV	SARS	Mouse/2 doses (~5 × 10^8^ pfu/mL)	Baculovirus act as adjuvant	s.c.	Production of high specific and neutralizing S protein antibodies.	[92]
Display A or P1 fused to GP64 under Pph	Site A (138-156aa) of VP1 from FMDV	HFMD	Mouse/2 doses (1 × 10^9^ pfu)	Freund’s adjuvant	i.p.	High specific antibodies and seroneutralizing titers. Shows protection in challenge study.	[40]
Parasite vaccines							
Display Pfs25 cloned between SP and TM of GP64 protein, under Pph.	Pfs25 (23-195aa) from *Plasmodium falciparum*	Malaria	Mouse/3 doses (1 × 10^8^ pfu)	Baculovirus act as adjuvant	i.n. and i.m.	High levels of Pfs25-specific antibodies. In passive immunization shows transmission-blocking effect (>90% reduction in infection intensity). In active immunization shows transmission blocking (83% i.n. and ~95% i.m.)	[93]
Display Pvs25 cloned between SP and TM of GP64 protein, under PCMV and Pph	Pvs25 (23-195aa) from *Plasmodium vivax*	Malaria	Mouse/4 doses (5 × 10^7^ pfu)	Baculovirus act as adjuvant	i.n. and i.m.	High Pvs25-specific antibody titers. Mixed Th1/Th2 response (IgG1, IgG2a and IgG2b). Transmission-blocking effect (94%) Active immunization shows transmission-blocking (92.1% i.n. and 83.8% i.m.)	[94]
Display PyMSP1_19_ protein cloned between SP and TM of GP64 protein under Pph.	PyMSP1_19_ from *Plasmodium yoelii*	Malaria	Mouse/3 doses (5 × 10^7^ pfu)	Baculovirus act as adjuvant	Oral, i.n. and i.m.	High titers of PyMSP1_19_ -specific antibodies (i.n. and i.m.). Mixed Th1/Th2 response (IgG1, IgG2a and IgG2b). Partial protection against lethal challenge.	[95]
Display MSP1_19_ or AMA1 cloned between SP and TM of GP64 protein under Pph.	MSP1_19_ from Pb, Pf and Py. AMA1 (domain I, II and III o only domain III) from Py.	Malaria	Mouse/3 doses (5 × 10^7^ pfu)	Baculovirus act as adjuvant	i.m. and i.n.	High level PfMSP1_19_-specifc antibody titers. High titers of PyAMA1-specifc antibodies. Partial protection in challenge study.	[96]
Display CSP protein cloned between SP and TM of GP64 protein	CSP from Pb	Malaria	Mouse/2 or 3 doses (1 × 10^8^ pfu)	Baculovirus act as adjuvant	i.m.	High levels and IFN-γ. Protection (60%) in challenge study.	[97]
Display CSP protein cloned between SP and TM of GP64 under PCMV and Pph promoters	CSP_(21-305)_ from Pb	Malaria	Mouse/3 doses (1 × 10^8^ pfu)	Baculovirus act as adjuvant	i.m.	High PbCSP-specific antibody titers. Mixed Th1/Th2 response (IgG1 and IgG2a). Specific CD8^+^ T-cell response. Partial parasitemia protection (~70%) in challenge study.	[98]
Display CSP protein under Pph and another copy under PCMV	CSP from Pf	Malaria	Mouse/2 doses Prime: ChAd63-PfCSP (5 × 10^7^ pfu) Boost: emBDES-PfCSP/IL12 (2 × 10^8^ pfu)	Baculovirus act as adjuvant	i.m.	Induces high anti-PfCSP IgG titers. High levels and IFN-γ. Increases memory CD8^+^ T-cell numbers. Protection in challenge study.	[99]
Display CSP and Pvs25 proteins cloned between SP of GP64 and TM of VSV-G under PCMV and Pph	CSP and Pvs25 from Pv	Malaria	Mouse/3 doses (1 × 10^8^ pfu)	Baculovirus act as adjuvant	i.m.	High CSP and vs25-specific antibody titers. Transmission-blocking effect (82%). Transmission-blocking in vivo (84%). Partial protection in challenge study.	[100]
Display CSP protein cloned between SP and TM of GP64 under Pph	CSP from Pf	Malaria	Mouse/2 doses (1 × 10^8^ pfu)	Baculovirus act as adjuvant	i.m.	High anti-CSP antibody titers. Mixed Th1/Th2 response (IgG1 and IgG2a). High levels and IFN-γ. Induces specific CD4^+^ and CD8^+^ T-cell responses.	[101]
Display CSP proteins cloned in C-terminus of SP of GP64 and N-terminus of TM GP64 or TM VSV-G under PCMV and Pph promoters	CSP from Pf	Malaria	Mouse/4 doses (1 × 10^8^ pfu)	Baculovirus act as adjuvant	i.m.	High antibody titers. Induces IgG1, IgG2a and IgG2b (Th1/Th2). Partial protection in challenge study.	[102]

**Abbreviations:** HA, hemagglutinin; NA, neuraminidase; ie1, immediate early promoter 1; WSSV, White Spot Syndrome Virus; pfu, plaque forming unit; rCTB; recombinant cholera toxin; HI, hemagglutination inhibition; GP64, envelope glycoprotein; AcMNPV, Autographa californica nuclear polyhedrosis virus; ED, ectodomain; SP, signal peptide; TM, transmembrane; CTD, cytoplasmic tail; Pph, polyhedrin promoter; Pp10, p10 promoter; HFMD, hand, foot and mouth disease; FMDV, foot-and-mouth disease virus; HEV71, human enterovirus 71; s.c., subcutaneous; i.m., intramuscular; i.p., intraperitoneal; i.n., intranasal; CPC-2, Canine parvovirus type 2; RVG, rabies virus glycoprotein; PCMV, early cytomegalovirus promoter; IFU, infection units; CyHV-2, cyprinid herpesvirus 2; VSV-G, glycoprotein of vesicular stomatitis virus; PCV2, porcine circovirus type 2; PMWS, post-weaning multisystemic wasting syndrome; Mhp, Mycoplasma hyopneumoniae; MPS mycoplasmal pneumonia of swine; GP5, glycoprotein 5; PRRS, porcine reproductive and respiratory syndrome; PRRSV, porcine reproductive and respiratory syndrome virus; E, envelop glycoprotein; ZIKV, zika virus; PEDV, porcine epidemic diarrhea virus; PED, porcine epidemic diarrhea; S, spike protein; S1, S1 domain of S protein; JEV, Japanese encephalitis virus; JE, Japanese encephalitis; IB, infectious bronchitis; IBV, avian infectious bronchitis; gD, glycoprotein D; BHV-1, bovine herpesvirus-1; E2 envelop glycoprotein E2; CSF, classical swine fever; CSFV classical swine fever virus; NS3, non-estructural protein 3; SARS, severe acute respiratory syndrome; SARS-CoV, severe acute respiratory syndrome-associated coronavirus; P1, precursor polyprotein; MSP1, merozoite surface protein 1, Pfs25, *Plasmodium falciparum* surface antigen 25; Pvs25, *Plasmodium vivax* surface antigen 25; PyMSP1_19_, 19-kDa *Plasmodium yoelii* merozoite surface protein 1; AMA1, apical membrane antigen 1; Pv, *Plasmodium vivax,* Pb, *Plasmodium berghei*; Py, *Plasmodium yoelii;* Pf, *Plasmodium falciparum*; PbCSP, *Plasmodium berghei* circumsporozoite protein; CSP, circumsporozoite.

## Data Availability

The data presented in this study is available upon request.

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
