# Peer review of "Baculovirus Display of Peptides and Proteins for Medical Applications"

_viruses, 2023, doi:10.3390/v15020411_

Round 1

Reviewer 1 Report

This is a timely review of a topic within baculovirus expression vectors that has not received as much attention as it deserves in recent years.  The use of baculovirus vectors for surface display of proteins has been around for quite a few years but is often ignored by reviewers in favour of the more mainstream applications of insect virus technology.

This review gives a good account of the field as well as including a little of the authors own work at the end of the paper. It is informative to all readers, irrespective of their prior knowledge.  Descriptions are included of the basis of baculovirus vectors in general so that anyone new to the area can understand how the technology works.  My only criticism of the Introduction is that it makes no mention of the occluded form of baculovirus particles, where enveloped nucleocapsids are packaged within polyhedra.  I know that this is not relevant to the topic being reviewed but it would help in the general education of the non-virologist to appreciate the natural form of the virus that is maintained in whole insects.

A further criticism concerns the two figures, which are very small in scale and difficult to decipher. These should be enlarged so that the lettering within boxes can be read more easily.  The Table 1 is also rather clumsy and would benefit from better formatting. Use of a smaller font may help in reducing the number of lines in text throughout.

Overall, this is a useful addition to the literature on baculovirus technologies. 

Author Response

REVIEWER 1

This is a timely review of a topic within baculovirus expression vectors that has not received as much attention as it deserves in recent years.  The use of baculovirus vectors for surface display of proteins has been around for quite a few years but is often ignored by reviewers in favor of the more mainstream applications of insect virus technology.

This review gives a good account of the field as well as including a little of the authors own work at the end of the paper. It is informative to all readers, irrespective of their prior knowledge.  Descriptions are included of the basis of baculovirus vectors in general so that anyone new to the area can understand how the technology works.  My only criticism of the Introduction is that it makes no mention of the occluded form of baculovirus particles, where enveloped nucleocapsids are packaged within polyhedra.  I know that this is not relevant to the topic being reviewed but it would help in the general education of the non-virologist to appreciate the natural form of the virus that is maintained in whole insects.

AUTHORS: We have included a line to address your concern. We explained in the revised version the two forms in which baculoviruses are produced (occluded and budded).

A further criticism concerns the two figures, which are very small in scale and difficult to decipher. These should be enlarged so that the lettering within boxes can be read more easily.  The Table 1 is also rather clumsy and would benefit from better formatting. Use of a smaller font may help in reducing the number of lines in text throughout.

AUTHORS: We appreciate your input, which most certainty will improve our work. In the initial submission figures had to fit the word format provided by the journal, however for the final version of our work figures will be submitted separately as high-resolution images. We have reduced the font size to improve the table readability.

Overall, this is a useful addition to the literature on baculovirus technologies. 

Reviewer 2 Report

The review by Rodríguez-Hernández et al. summarized the progress of baculovirus display, particularly the BV display platform using GP64, and highlighted the potential to display peptides based on the authors’ successful trial to display a 9 aa Bk on BV.

General comments:

Overall, the advances on BV display were introduced comprehensively, and the classical paper in the field were well included. However, much grammar errors should be corrected and figures should be tuned before being published in Viruses.

Below are editorial suggestions or minor concerns:

1.    There are a number of grammar errors throughout the text, here are several examples:

Line 37, Baculoviruses collect

Line 66, Sf21

Line 154, but most recently

Line 157, proven successful

Line 209, signal peptide (SP)

Line 235, be replaced by

Line 271, mediated by

Line 383, there are three main

Line 411, there are many

2.    Both figure 1 and 2 were suggested to be enlarged.

3.    Please tune fig. 2A to show labels clearly.

4.    As there are no method details for Fig.2, I wonder whether purified Bac-Bk BV virions were used for calcium signal detection, instead of BV supernatant harvested from cell culture. And do the authors have evidence to claim the chimeric GP64-Bk was indeed localized on the virion surface? Any influence of such chimeric GP64 on BV infectivity?

Author Response

REVIEWER 2

The review by Rodríguez-Hernández et al. summarized the progress of baculovirus display, particularly the BV display platform using GP64, and highlighted the potential to display peptides based on the authors’ successful trial to display a 9 aa Bk on BV.

General comments:

Overall, the advances on BV display were introduced comprehensively, and the classical paper in the field were well included. However, much grammar errors should be corrected, and figures should be tuned before being published in Viruses.

AUTHORS: We appreciate the time devoted to analyzing our manuscript. We have checked the entire manuscript for grammatical errors and misspellings. Unfortunately including the figures in a single word file is not optimal for figure display, although this is a journal requirement. We have prepared high-resolution images for the final submission.

Below are editorial suggestions or minor concerns:

  1. There are a number of grammar errors throughout the text, here are several examples:

Line 37, Baculoviruses collect

Line 66, Sf21

Line 154, but most recently

Line 157, proven successful

Line 209, signal peptide (SP)

Line 235, be replaced by

Line 271, mediated by

Line 383, there are three main

Line 411, there are many

AUTHORS: We have checked the entire manuscript for grammatical errors and misspellings.

  1. Both figure 1 and 2 were suggested to be enlarged.
  2. Please tune fig. 2A to show labels clearly.

AUTHORS: We will submit high-resolution images of these figures in the next submission. Adding the figure to the word document altered the position of text in the figure.

  1. As there are no method details for Fig.2, I wonder whether purified Bac-Bk BV virions were used for calcium signal detection, instead of BV supernatant harvested from cell culture. And do the authors have evidence to claim the chimeric GP64-Bk was indeed localized on the virion surface? Any influence of such chimeric GP64 on BV infectivity?

AUTHORS: We apologize for this omission. Indeed, we have used purified baculoviruses using a typical centrifugation protocol (reference added). We cannot detect a small peptide, such as bradykinin, because the lack of a specific antibody. We detected GP64-GFP on the surface of baculoviruses and by western blot since wild type GP64 and GP64-GFP have different molecular weights. All recombinant baculoviruses carried an extra copy of GP64 (GP64-GFP and GP64-BK). We did not find any differences in infectivity (MOI) of recombinant GP64-GFP, GP64-BK when compared to baculovirus that did not carry an additional copy of GP64.

Reviewer 3 Report

The baculovirus display system is an upcoming technology for the presentation of recombinant proteins and peptides on baculovirus particles. The display is mediated via fusion to the baculovirus membrane or capsid proteins. Baculovirus display technology has several applications, including diagnostics, antigen carrier and gene delivery. The authors provide an overview of the baculovirus expression vector system, modes of protein display, advantages and possible applications. Finally, the authors show novel data on the display of a peptide from bradykinin on recombinant baculoviruses by fusion to GP64.

The baculovirus protein display technology has been reviewed extensively in 2020 by Tsai et al. (doi:10.21775/cimb.034.231). In recent years the application and developments within this technology have probably expanded. The authors have included some novel research but it would be useful to include more recent and original articles to provide for a complete and informative overview of the technology including recent advances.

Section 1.1 (Characteristics of the baculovirus expression vector system) is an extensive overview of the different commercial products available for recombinant baculovirus construction. Could the authors please elaborate on the importance of these techniques for the baculovirus display technology? What considerations should be taken for the best virus yield or most efficient protein display? This section could otherwise be shortened to summarize the most important platforms.  

The authors frequently refer to previous research studies without including adequate references. This should be addressed at: ‘in these studies’ (line 184), ‘several studies’ (line 189), ‘several studies’ (line 190), ‘several research papers’ (line 208), ‘several strategies’ (line 217), ‘most of the studies’ (line 226), and ‘in previous studies’ (line 231). Please include an adequate number of references to make these statements.

The authors finalize their review paper by showing successful baculovirus display of a bioactive peptide of bradykinin. Interestingly, this peptide retains its activity when fused to the N-terminal sequence of GP64. It would be useful to include methodological details and elaborate on the experimental set-up. For example, does the baculovirus encode a non-fused (wildtype) copy of GP64? How are the experiments performed? Which insect cell line was used? Etc.

In conclusion, the authors provided an overview of the baculovirus display technology and show novel results on the display of bioactive bradykinin. It is questionable if the review contains enough novel research and advances.

Minor comments

-       Line 23, the full name of the virus should be mentioned here (Autographa californica multiple nucleopolyhedrovirus)

-       Some spelling errors at ‘Nucleopolyhedrovirus’ (Line 59), ‘most’ (line 154), ‘by’ (line 271), ‘available’ (line 450), and GP64 (GPGP64, table 1).

-       Figure 1 and Figure 2, the text in the figure has shifted and does not line up to the boxes anymore. The general readability of the figure can also be improved by choosing a larger font size for the text.

Author Response

REVIEWER 3

The baculovirus display system is an upcoming technology for the presentation of recombinant proteins and peptides on baculovirus particles. The display is mediated via fusion to the baculovirus membrane or capsid proteins. Baculovirus display technology has several applications, including diagnostics, antigen carrier and gene delivery. The authors provide an overview of the baculovirus expression vector system, modes of protein display, advantages and possible applications. Finally, the authors show novel data on the display of a peptide from bradykinin on recombinant baculoviruses by fusion to GP64.

The baculovirus protein display technology has been reviewed extensively in 2020 by Tsai et al. (doi:10.21775/cimb.034.231). In recent years the application and developments within this technology have probably expanded. The authors have included some novel research but it would be useful to include more recent and original articles to provide for a complete and informative overview of the technology including recent advances.

AUTHORS: We would like to thank you for the time devoted to reading our manuscript. Following your pertinent suggestion, we have included more recent literature, including the work by Tsai et al.

Section 1.1 (Characteristics of the baculovirus expression vector system) is an extensive overview of the different commercial products available for recombinant baculovirus construction. Could the authors please elaborate on the importance of these techniques for the baculovirus display technology? What considerations should be taken for the best virus yield or most efficient protein display? This section could otherwise be shortened to summarize the most important platforms.  

AUTHORS: We have added a few lines in this section to discuss the relevance of new baculovirus expression systems for baculovirus display.

The authors frequently refer to previous research studies without including adequate references. This should be addressed at: ‘in these studies’ (line 184), ‘several studies’ (line 189), ‘several studies’ (line 190), ‘several research papers’ (line 208), ‘several strategies’ (line 217), ‘most of the studies’ (line 226), and ‘in previous studies’ (line 231). Please include an adequate number of references to make these statements.

AUTHORS: We have added the references in the lines indicated.

The authors finalize their review paper by showing successful baculovirus display of a bioactive peptide of bradykinin. Interestingly, this peptide retains its activity when fused to the N-terminal sequence of GP64. It would be useful to include methodological details and elaborate on the experimental set-up. For example, does the baculovirus encode a non-fused (wildtype) copy of GP64? How are the experiments performed? Which insect cell line was used? Etc.

AUTHORS: Following your pertinent suggestion we have included in the figure legend more methodological details. All GP64 fusion proteins used in this study were included as a second copy drivne by the polyhedrin promoter, as shown in figure 2A.

In conclusion, the authors provided an overview of the baculovirus display technology and show novel results on the display of bioactive bradykinin. It is questionable if the review contains enough novel research and advances.

Minor comments

-       Line 23, the full name of the virus should be mentioned here (Autographa californica multiple nucleopolyhedrovirus)

AUTHORS: Full name included.

-       Some spelling errors at ‘Nucleopolyhedrovirus’ (Line 59), ‘most’ (line 154), ‘by’ (line 271), ‘available’ (line 450), and GP64 (GPGP64, table 1).

AUTHORS: We have checked the entire manuscript for grammatical errors and misspellings.

-       Figure 1 and Figure 2, the text in the figure has shifted and does not line up to the boxes anymore. The general readability of the figure can also be improved by choosing a larger font size for the text.

AUTHORS: In the initial submission figures had to fit the word format provided by the journal, however for the final version of our work figures will be submitted separately as high-resolution images. In post-production the journal will accommodate the figures and table for better readability.

Round 2

Reviewer 3 Report

I mainly looked at the added sections, but also found some issues in earlier written parts. 

L36-40 is thin on references.

L48 VP39 (the major capsid protein) does not surround the entire virus capsid, but is the major component of the nucelocapsid, as they already suggest between brackets.

L50-52 This statement: After baculovirus capsid is assembled, genomic DNA is transported into 50 the capsid using an energy-dependent process, similarly to what other large DNA genome 51 viruses do, such as herpes viruses.is incorrect. Baculoviruses do not first make an empty capsid, they also have a completely different form as herpesvirus capsids.

L54 produced of the protein polyhedrin.: mainly consisting of ....

L55: what do they mean here by prorogation? suggest to replace by viral phenotype

L56: it is not the virions that are transported to the cell surface for budding, but the  nucleocapsids

L59: retaining their infectivity.

L67: genus Alphabaculovirus

L196: have been displayed

L377: Is adeno-associated viruses used to display proteins? I know them as gene therapy vecors, but if so please add reference. 

Please check the whole paper carefully for small inconsistencies and grammar issues. I also do not think the the journal normally capitalizes the first words of all the nouns in titles in teh references. Please carefully check.

Author Response

Comments and Suggestions for Authors

I mainly looked at the added sections, but also found some issues in earlier written parts. 

L36-40 is thin on references.

AUTHORS: We added a couple more.

L48 VP39 (the major capsid protein) does not surround the entire virus capsid, but is the major component of the nucelocapsid, as they already suggest between brackets.

AUTHORS: Changed to: VP39 is the major capsid protein, while GP64 concentrates in one pole of the rod.

L50-52 This statement: After baculovirus capsid is assembled, genomic DNA is transported into 50 the capsid using an energy-dependent process, similarly to what other large DNA genome 51 viruses do, such as herpes viruses.is incorrect. Baculoviruses do not first make an empty capsid, they also have a completely different form as herpesvirus capsids.

AUTHORS. This sentence has been removed from the revised version.

L54 produced of the protein polyhedrin.: mainly consisting of ....

AUTHORS: Changed to: known as polyhedra mainly consisting of the protein polyhedrin.

L55: what do they mean here by prorogation? suggest to replace by viral phenotype

AUTHORS: Changed to viral phenotype as suggested.

L56: it is not the virions that are transported to the cell surface for budding, but the  nucleocapsids

AUTHORS: Thank you for the clarification. This line was redundant and we removed it from the revised version.

L59: retaining their infectivity.

AUTHORS: Corrected. Thank you.

L67: genus Alphabaculovirus

AUTHORS: Corrected. Thank you.

L196: have been displayed

AUTHORS: Corrected. Thank you.

L377: Is adeno-associated viruses used to display proteins? I know them as gene therapy vectors, but if so please add reference. 

AUTHORS: There are a few reports of adeno-associated virus display (Molecular Therapy Volume 21, Issue 1, January 2013, Pages 109-118) and (ACS Synth. Biol. 2020, 9, 2246−2251). We have included a couple of references about it.

Please check the whole paper carefully for small inconsistencies and grammar issues. I also do not think the the journal normally capitalizes the first words of all the nouns in titles in teh references. Please carefully check.

AUTHORS: Following your pertinent suggestion, we have checked the entire manuscript for grammatical errors. Regarding the format of the references, we used the style template provided by the journal and Mendeley automatically arranged the references. We will check with the journal during production if our manuscript is accepted for publication. Thank you.